# Robust Cherry Tomatoes Detection Algorithm in Greenhouse Scene Based on SSD

**Ting Yuan \*, Lin Lv, Fan Zhang, Jun Fu, Jin Gao, Junxiong Zhang, Wei Li, Chunlong Zhang and Wenqiang Zhang**

College of Engineering, China Agricultural University, Qinghua a Rd.(E)No.17, Haidian District,
Beijing 100083, China; llin8090@163.om (L.L.); zhangfancau@foxmail.com (F.Z.); fj163123@163.com (J.F.);
gaojincau@foxmail.com (J.G.); cau007@cau.edu.cn (J.Z.); liww@cau.edu.cn (W.L.); zcl1515@cau.edu.cn (C.Z.);
zhangwq@cau.edu.cn (W.Z.)
\* Correspondence: yuanting122@hotmail.com; Tel.: +86-138-1059-2156

**Abstract:** The detection of cherry tomatoes in greenhouse scene is of great significance for robotic harvesting. This paper states a method based on deep learning for cherry tomatoes detection to reduce the influence of illumination, growth difference, and occlusion. In view of such greenhouse operating environment and accuracy of deep learning, Single Shot multi-box Detector (SSD) was selected because of its excellent anti-interference ability and self-taught from datasets. The first step is to build datasets containing various conditions in greenhouse. According to the characteristics of cherry tomatoes, the image samples with illumination change, images rotation and noise enhancement were used to expand the datasets. Then training datasets were used to train and construct network model. To study the effect of base network and the input size of networks, one contrast experiment was designed on different base networks of VGG16, MobileNet, Inception V2 networks, and the other contrast experiment was conducted on changing the network input image size of 300 pixels by 300 pixels, 512 pixels by 512 pixels. Through the analysis of the experimental results, it is found that the Inception V2 network is the best base network with the average precision of 98.85% in greenhouse environment. Compared with other detection methods, this method shows substantial improvement in cherry tomatoes detection.

**Keywords:** cherry tomatoes; deep learning; SSD; robotic harvesting

## 1. Introduction

China is the world's largest tomato production and consumption country. The area of tomato plantation is about 1.0538 million hm$^2$ in an average year, with a total production of 54.133 million Mg, which accounts for 21% of the world's total production [1]. Cherry tomatoes are one of the favorite tomato varieties to be planted. It is all attributed to the special flavor, easy planting, and high economic benefits [2]. In China, tomato harvesting manily relies on manual labor, and the labor cost accounts for about 50~60% of the total cost of tomato production [3]. The rapid development and widespread use of harvesting robots are due to their high harvesting efficiencies and low maintenance costs. However, it is extremely difficult to develop a vision system as intelligent as humans for detection. There are plenty of problems to be solved, such as uneven illumination in unstructured environment, fruit occlusion by the main stem, overlapping fruit and other unpredictable factors. Therefore, the study of cherry tomatoes detection has a crucial practical significance to enhance positioning accuracy and environmental adaptability of the harvesting robots.

Since 1990, a great effort has been made in visual recognition technology for picking robots around the world. A series of traditional fruit detection and recognition algorithms were successfully

proposed. Kondo, Zhang, Yin, et al. [4–6] used the difference of color feature between mature tomatoes and branches or leaves to identify mature tomatoes. By using threshold segmentation, it achieved an accuracy rate of 97.5% on average, but it was failed to detect immature tomatoes. Zhao, Feng et al. [7,8] proposed the mature tomato segmentation methods based on color space,such as HIS and Lab space. However, these algorithms were heavily rely on the effectiveness of threshold segmentation method and color space [9].

More and more research is focused on using machine learning or deep learning to solve the problem of fruit recognition.Wang et al. [10] used a color-based K-means clustering algorithm to segment litchi images with an accuracy rate of 97.5%. However, complex background noise and growth differences had a profound impact on object detection. Krizhevsky et al. [11] proposed AlexNet convolutional neural network (CNN [12]) architecture, which avioded the complicated feature engineering and reduced the requirements of image preprocessing. CNN behaved better than traditional methods in ability to learn features. Zhou et al. [13] made structural adjustments based on the VGGNet convolutional neural network, which could be classified into ten tomato organs and the classification error rates were all lower than 6.392%. Fu et al. [14] stated a deep learning model based on LeNet convolutional neural network for multi-cluster kiwi fruit image detection, and the accuracy rate achieved 89.29%. Peng et al. [15] adopted an improved SSD model to make a multi-fruit detection system with an average detection accuracy of 89.53%.

SSD could capture the information of an object and is anti-interference in real case. It is an end-to-end object detection framework can directly complete the location task and classification task in one step. Therefore, the detection accuracy is higher than other frameworks which relied on candidate regions. However, there is poorly research on fruit detection using SSD. Therefore, considering different lighting conditions and various tomato growth states, an improved model for cherry tomatoes detection based on the SSD proposed by Liu et al. [16] is presented in this paper. The model is improved by using end-to-end training method to adaptively learn the characteristics of cherry tomatoes in different situations, which achieves fruit recognition and localization in unstructured harvesting environment.

## 2. Data Material

### 2.1. Image Acquisition

Images of cherry tomatoes were taken from in April and July 2019 in Hongfu Agricultural Tomato Production Park in Daxing District, Beijing, China. The image acquisition device was an Intel RealSense D435 depth camera. The image format is JPG, and the image resolution is 1024 × 960 pixels. The depth camera was placed on a tripod placed on a railcar between cherry tomatoes plants. The distance between the image acquisition device and the tomato plants was about 600mm, which is in accordance with the best operation distance for robotic harvesting. In order to ensure the diversity of image samples and take into account the variability of light and the difference of fruit strings, a total of 1730 cherry tomatoes images were collected at different times of the sunny days and the cloudy days.

Figure 1 is some image samples of cherry tomatoes under various environments such as different lighting conditions, different growth stages, stacking conditions, and occlusion. The image samples show that the color and shape characteristics of cherry tomatoes vary according to different factors. The color characteristics change greatly under different lighting conditions and different tomato growth stages. The shape characteristics are dramatically impacted by different shooting angles of the depth camera. One image sample may simultaneously include several conditions (such as changes in illumination, overlapped or blocked by the main stem and so on). Therefore, the method of detecting cherry tomatoes based on color and shape features has certain limitations. It is extremely difficult to extract the complete cherry tomatoes features. An automatic feature extraction network was considered to gain cherry tomatoes features. And an end-to-end network training method was used to improve the adaptive resistance of the neural network.

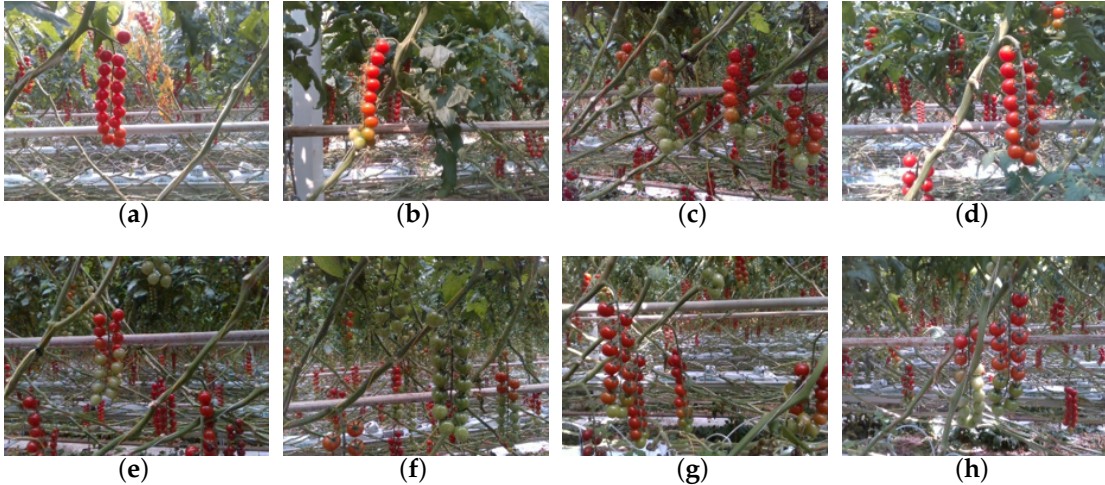

**Figure 1.** Some examples of cherry tomatoes in different conditions: (**a**) backlight conditions, (**b**) light conditions, (**c**) shaded conditions, (**d**) ripe conditions, (**e**) half-ripe conditions, (**f**) immature conditions, (**g**) overlapped conditions, and (**h**) occlusion by main stem conditions.

## 2.2. Sample Data Set

To make the model more robust, data augmentation techniques for all the number of original images were applied with the methods of rotating, brightness adjustment and noising. Considering the randomness of growth stages of cherry tomatoes, images were augmented with rotations of $\pm 5°$, $\pm 10°$. After data augmentation, the number of image samples was expanded to 3460. Examples of data augment images are shown in Figure 2.

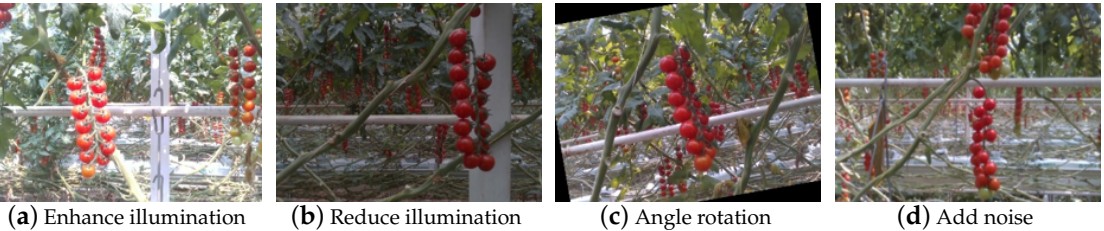

(**a**) Enhance illumination　(**b**) Reduce illumination　(**c**) Angle rotation　(**d**) Add noise

**Figure 2.** Images of cherry tomatoes after data augmentation: (**a**) after illumination enhancement, (**b**) after illumination reducement, (**c**) after angle rotation and (**d**) after add noise.

All samples are divided into training set, test set, and validation set as shown in Table 1 according to the PASCAL VOC2007 [17] dataset. The training and test sample sets are used for training and evaluating models respectively. And the validation set is employed to optimize hyperparameters during model training. The ratio of the three sample sets is about 8:1:1. All datasets were randomly selected, which not only ensured the equal distribution of the sample set but also met the reliability of the evaluation.

**Table 1.** Quantity of sample data sets.

| Research Object | Data Sets | | | |
| --- | --- | --- | --- | --- |
| | Training Set | Test Set | Validation Set | Total Quantity |
| Cherry tomato | 2768 | 346 | 346 | 3460 |

## 3. Theoretical Background

### 3.1. Classical SSD Deep Learning Model

The methods of deep convolutional neural networks in the field of target detection are mainly divided into two categories. One is the target detection framework based on regional proposals, including RCNN [18], Fast R-CNN [19], Faster R-CNN [20], and FPN [21], etc. The other is a framework without candidate regions, the landmark algorithms of which are YOLO [22], SSD. SSD is able to predict the location and classification in one step. Compared with frameworks relied on candidate regions, SSD not only ensures detection accuracy but also improves detection speed. In addition, SSD uses multiple feature layers of different sizes to predict the regression box. Although the speed of SSD is slightly slower, the detection accuracy is significantly improved. Considering the complexity, accuracy, and speed requirements of cherry tomatoes detection, the model based on the SSD was used.

The structure of the SSD model is shown in Figure 3. VGG16 was used as the base network of SSD. Several layers were added to the end of the base network. During training, image and the corresponding Ground-Truth Box were inputted at the same time. A default box set and the confidence of object category in box were subsequently generated by each feature map in 6 layers.

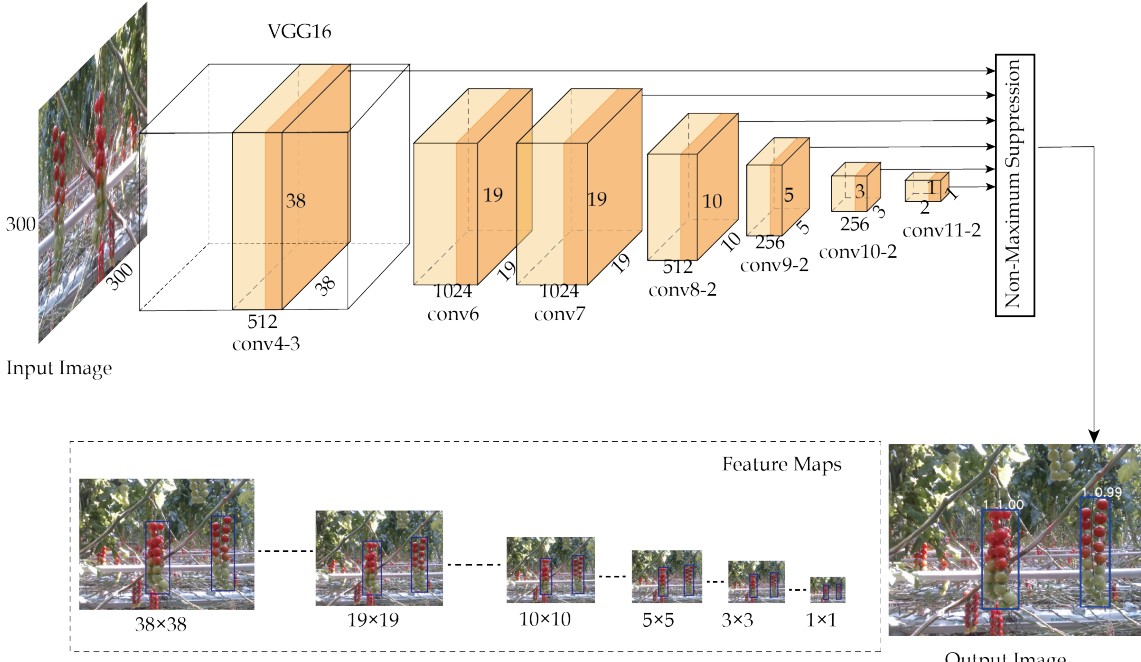

**Figure 3.** Architecture of classical SSD deep learning networks.

Use $x_{ij}^p = \{1, 0\}$ as an indicator for matching the *i*-th default box to the *j*-th ground truth box of category *p*. When $x_{ij}^p \geq 1$, it means that there are more than one default box match the *j*-th ground truth box. The overall loss function is expressed by the weighted sum of the confidence loss ($L_{conf}$) and the localization loss ($L_{loc}$). The equation is shown in Equation (1):

$$L(x, c, l, g) = \frac{1}{N} \left( L_{conf}(x, c) + \alpha L_{loc}(x, l, g) \right) \tag{1}$$

In Equation (1), *N* is the number of matched default boxes. *x* and *c* represent the true value of the category and the predicted value of the confidence of the category respectively. *α* is a weighting factor and is set to 1 by cross validation. The confidence loss is the softmax loss over multiple classes

confidences. It can be expressed as shown Equation (2), where $c_i^p$ is the confidence of the $i$-th bounding box of the category $p$.

$$L_{conf}(x,c) = -\sum_{i \in Pos}^{N} x_{ij}^p \log\left(\hat{c}_i^p\right) - \sum_{i \in Neg} \log\left(\hat{c}_i^0\right) \quad \text{where } \hat{c}_i^p = \frac{\exp\left(c_i^p\right)}{\sum_p \exp\left(c_i^p\right)} \tag{2}$$

$L_{loc}$ is a Smooth L1 loss [19] between the predicted box ($l$) and the ground truth box ($g$), which can be expressed as shown in Equation (3):

$$L_{loc}(x,l,g) = \sum_{i \in Pos} \sum_{m \in \{cx,cy,w,h\}} x_{ij}^k \, \text{smooth}_{L1}\left(l_i^m - \hat{g}_j^m\right) \tag{3}$$

where $\{cx, cy, w, h\}$ are the center coordinates, width and height of the predicted box. The translations $d_i^{cx}$, $d_i^{cy}$ and scale scaling factors $d_i^w$, $d_i^h$ are used to get approximate regression prediction boxes of real labels $\hat{g}_j^w$, shown as Equation (4):

$$\begin{aligned}
\hat{g}_j^{cx} &= \frac{g_j^{cx} - d_i^{cx}}{d_i^w} & \hat{g}_j^{cy} &= \frac{g_j^{cy} - d_i^{cy}}{d_i^h} \\
\hat{g}_j^{w} &= \log\left(\frac{g_j^w}{d_i^w}\right) & \hat{g}_j^{h} &= \log\left(\frac{g_j^h}{d_i^h}\right)
\end{aligned} \tag{4}$$

### 3.2. Improved SSD Deep Learning Model

Simonyan [23] supported that the recognition accuracy will increase with the deepening of the network. However, the model did not have a significant improvement on detection speed when the accuracy was improved. Moreover, a small sized and portable robot is too limited to support the complexity detection task. Although several options are available to solve this obstacle, the best one seems to be using lightweight network as base network of SSD. The use of lightweight network models such as MobileNet [24] and Inception V2 [25] can effectively improve the real time performance of object detection without reducing the detection accuracy.

SSD_MobileNet model and SSD_Inception V2 model use MobileNet and Inception V2 networks instead of VGG16 network as the base network structure respectively. The main feature of MobileNet is that using depthwise separable convolutions to replace the standard convolutions of traditional network structures. Its significant advantages are high computational efficiency and small parameters of convolutional networks. The main feature of Inception V2 is to increase the complexity of the network by increasing the width of the network. And BN layer is used to standardize the input layer information. In general, Inception V2 lessen the matter of "Internal Covariate Shift", speeded up training and prevented the issue of gradient disappearing during training. At the same time, parameters, memory and computing resources are much lower than traditional networks.

Figure 4 shows the network structure of SSD_MobileNet. Based on the SSD network structure, convolutional layers are added to the end of the truncated base network. 6 layers of SSD_MobileNet network are extracted for object localization and classification prediction. Non-maximum Suppression(NMS) [26] is used to filter out repeated prediction boxes. The network structure of SSD_Inception V2 is similar to SSD_MobileNet. It contains 6 different layers like Mixed 4c layer, Mixed 5c layer, and other 4 layers with amount of channels 512, 256, 256, 128 respectively for prediction.

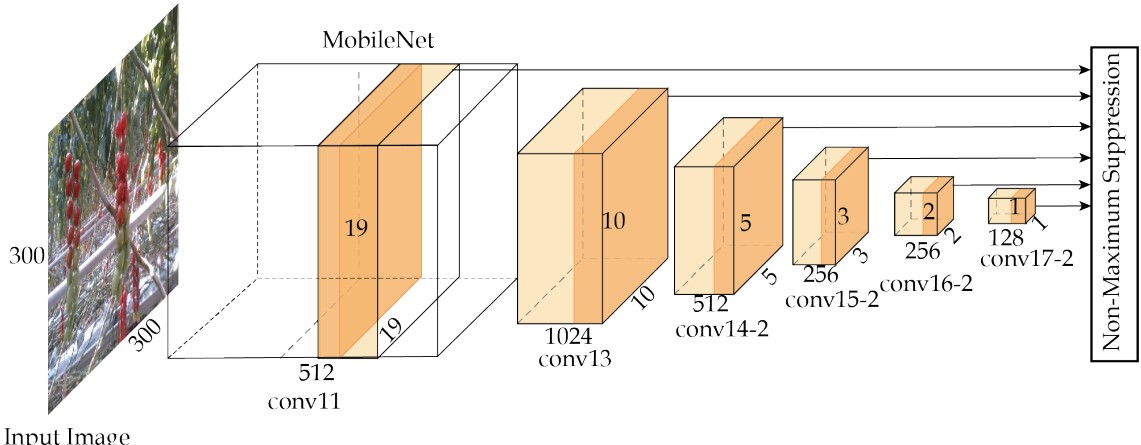

**Figure 4.** Architecture of improved SSD deep learning networks.

### 3.3. Overview of Detection Algorithm

The process of SSD-based detection can be summarized in the following steps: firstly, use the maximum of category confidence of each default box as its category and confidence. Next, filter out the predicted boxes with the category as the background, and filter out the low confidence predicted boxes according to the given threshold of confidence value. Thirdly, decode the default boxes of feature layers of different scales. Finally, apply NMS method to filter out the predicted boxes, and generate final prediction boxes.

## 4. Results and Disscussion

### 4.1. Experimental Setup

All experiments were performed on the Tensorflow framework with Intel$^{(R)}$ Core$^{TM}$ i7-8750H CPU@4.10GHz, 16GB memory, 1TB HDD + 256GB solid-state hard disk, NVIDIA GTX1070GPU, and the operating environment was Window 10(64 -bit) system, Python version 3.5.4, CUDA version 9.0.176 parallel computing architecture and cuDNN version 8.0 deep neural network library.

### 4.2. Experiment Design

The experimental sample images include various growth status and lighting conditions of cherry tomatoes in the greenhouse scene, such as sunny or uneven illumination and separated, occlusion, shadow, side-grown, and overlapped cherry tomatoes. The separated cherry tomatoes mean that fruit are independent and without any obstruction. The number of such cherry tomatoes accounts for more than 85% of the total number, which will greatly affect the detection accuracy rate of cherry tomatoes. The occlusion cherry tomatoes means tomatoes are obscured by the main stems, while the overlapped cherry tomatoes mean two or more bunches of tomatoes stacked on each other in the image. Some cherry tomatoes can be covered more than 50% of the whole bunch area, or several conditions arise in one bunch of cherry tomatoes, which increase the difficulty of tomato detection. In the comparative experiments, the sample images were divided into six categories: separated, occlusion, sunny or uneven illumination, shadow, side-grown and overlapped.

Four different SSD convolutional neural network models were compared to verify their performance in this paper. Firstly, different feature extractors of the SSD classical network were applied by changing network parameters and network depths, which directly affect detection time and effect of cherry tomatoes. In the tests, VGG16, MobileNet and Inception V2 were used as feature extractors to be discussed. Secondly, since the size of the network input images was sensitive to the small

detection objects, the networks of SSD300 and SSD512 with the input image sizes of $300 \times 300$ pixels and $512 \times 512$ pixels were selected respectively in the tests.

### 4.2.1. Experiment Parameters

To reduce training time and save training resources, transfer learning was used to train the training sample set based on COCO dataset. SSD300 and SSD512 networks used Adam as gradient descent algorithm, while SSD_MobileNet and SSD_Inception V2 networks used RMS props as gradient descent algorithm. The batch size was set to 4. The momentum and the weight decay were set to 0.9 and 0.0005 respectively. Used $10^{-4}$ learning rate for 70 K iterations and continue training for 10 K iterations with $10^{-5}$.

### 4.2.2. Evaluation Standard

Three indexes were applied to evaluate the performance of the proposed algorithm and developed algorithms: precision (P), false negative rate (F), and Intersection over Union (IOU). Among them, precision and false negative rate are used as evaluation index under different conditions in the test set, and IOU are used as the evaluation index of the detection result of a single object.

In the case that the default box was determined, the default box with the largest IOU and the default box which IOU is greater than 0.5 are defined as positive sample. False positive means that the samples that are actually positive samples but divided into negative samples. Precision refers to the proportion of positive samples of predicted samples among all positive samples, as an evaluation of the accuracy of the detection result. False negative rate refers to the proportion of misclassified positive samples of predicted samples among all positive samples. Precision and false negative rate are defined by Equations (5) and (6):

$$P = \frac{T_P}{T_P + F_P} \times 100\% \tag{5}$$

$$F = \frac{F_N}{T_P + F_N} \times 100\% \tag{6}$$

where $T_P$, $F_P$, $F_N$ are truth positive, false positive, false negative, respectively.

IOU refers to the ratio of the intersection to union of the area between the predicted box and the ground truth box as a symbol of the accuracy of the target location.

AP (average precision) as an evaluation index of detection precision on the entire test dataset, which is a criterion for evaluating the sensitivity of network to object. AP value is related to the precision and recall rate(R). The recall rate represents the proportion of the predicted correct boxes in all ground truth boxes, that means the completeness of a result. The precision and recall rate are defined by Equations (5) and (7):

$$R = \frac{T_P}{T_P + F_N} \times 100\% \tag{7}$$

AP is calculated by the integral of precision-recall curve. The higher the value of AP, the better the model performs. AP is defined by Equation (8):

$$AP = \int_0^1 PR\,dR \tag{8}$$

### 4.3. Experiment Results Analysis

The performance of the proposed four network models (SSD300, SSD512, SSD_MobileNet, and SSD_Inception V2) were evaluated using 50 bunches of randomly selected separated cherry tomatoes. Just test set was used, and the results are shown in Table 2. The precision of the test set were 88%, 92%, 92%, and 94% respectively, which shows that SSD_Inception V2 model is more effective for separated cherry tomato detection. Some image examples of the results are shown in Figure 5.

Because of integrated features, all the tomatoes were correctly detected under the separated conditions as expected. The IOU of four different models were 0.896, 0.865, 0.853 and 0.972 respectively. It can be observed that the SSD_Inception V2 model not only has the largest IOU, but also has the highest detection accuracy.

**Table 2.** The detection results of separated cherry tomatoes.

| Conditions | Methods | Tomato Amount | Correctly Identified Amount | P | Missed Amount | F |
|---|---|---|---|---|---|---|
| Separated | SSD300 | 50 | 44 | 88% | 6 | 12% |
| | SSD512 | | 46 | 92% | 4 | 8% |
| | SSD_MobileNet | | 46 | 92% | 4 | 8% |
| | SSD_Inception V2 | | 47 | 94% | 3 | 6% |

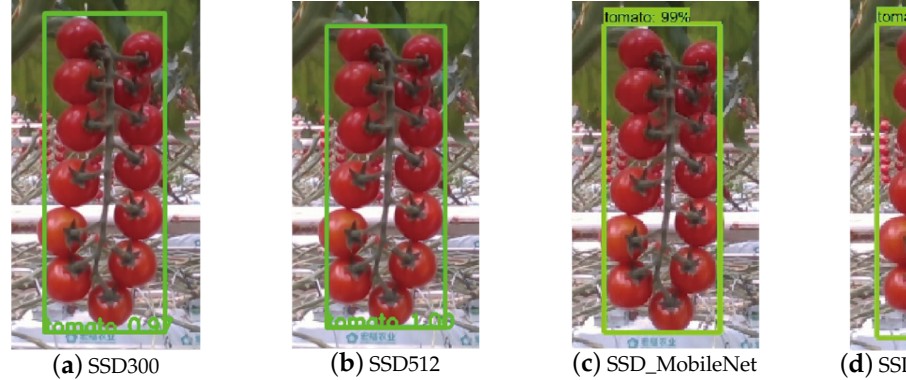

(**a**) SSD300　　(**b**) SSD512　　(**c**) SSD_MobileNet　　(**d**) SSD_Inception V2

**Figure 5.** The detection results of separated cherry tomatoes. (**a**) the result of SSD300 model, (**b**) the result of SSD512 model, (**c**) the result of SSD_MobileNet model and (**d**) the result of SSD_Inception V2 model.

Some cherry tomatoes are obscured by the main stems, which visually causes the target to be divided into two parts. The performance of the proposed network models were evaluated using 50 bunches of randomly selected cherry tomatoes covered by the main stem. The detection results are shown in Table 3. The precision of the test set were 60% , 90% , 94% , and 92%, respectively. Examples of cherry tomatoes obscured by the main stem are shown in Figure 6. In general, although the area covered by the main stem is less than 10% of the total cherry tomatoes area, it is easy to be missed or failed due to the interference of the main stem. The main causes of missed detection or wrong detection were that the severe deformation of the cherry tomatoes and the obstruction cherry tomatoes mistakenly identified as two separated bunches.

**Table 3.** The detection results of different methods under main stem occlusion condition.

| Conditions | Methods | Amount | Correctly Identified Amount | P | Missed Amount | F |
|---|---|---|---|---|---|---|
| Separated | SSD300 | 50 | 30 | 60% | 20 | 40% |
| | SSD512 | | 45 | 90% | 5 | 10% |
| | SSD_MobileNet | | 47 | 94% | 3 | 6% |
| | SSD_Inception V2 | | 46 | 92% | 4 | 8% |

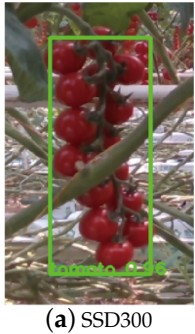 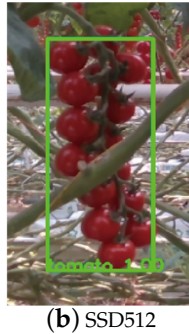 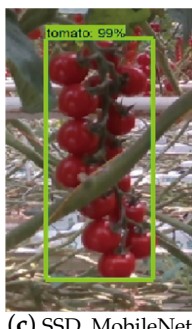 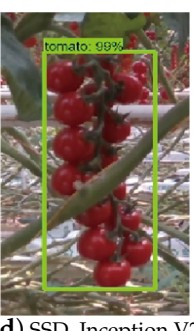

(**a**) SSD300      (**b**) SSD512      (**c**) SSD_MobileNet      (**d**) SSD_Inception V2

**Figure 6.** The detection results of obscured cherry tomatoes. (**a**) the result of SSD300 model, (**b**) the result of SSD512 model, (**c**) the result of SSD_MobileNet model and (**d**) the result of SSD_Inception V2 model.

Halations on the fruit surface which were caused by uneven illumination seriously affect tomato detection. 50 bunches of cherry tomatoes in direct sunlight or uneven light conditions and 50 bunches in shaded conditions were tested by the proposed four models. The results are shown in Table 4. The precision of the test set under direct sunlight or uneven light conditions were 40%, 90%, 92%, and 96% respectively. Relatively the precision of the test set under shadow conditions were 80%, 96%, 96% and 98% respectively. The test results proved that the three models except SSD300 were insensitive to illumination variation in the greenhouse environment. This is mainly due to excellent feature extraction function of network and data augmentation.

**Table 4.** The detection results of different methods under different lighting conditions.

| Conditions | Methods | Amount | Correctly Identified Amount | P | Missed Amount | F |
|---|---|---|---|---|---|---|
| Sunny or Uneven Illumination | SSD300 | 50 | 20 | 40% | 30 | 60% |
| | SSD512 | | 45 | 90% | 5 | 10% |
| | SSD_MobileNet | | 46 | 92% | 4 | 8% |
| | SSD_Inception V2 | | 48 | 96% | 2 | 4% |
| Shadow | SSD300 | 50 | 40 | 80% | 10 | 20% |
| | SSD512 | | 48 | 96% | 2 | 4% |
| | SSD_MobileNet | | 48 | 96% | 2 | 4% |
| | SSD_Inception V2 | | 49 | 98% | 1 | 2% |

Some examples of cherry tomatoes with uneven illumination are shown in Figure 7. Some fruit surfaces have white spots due to excessively strong illumination. Except for the SSD300 missed detection, the other three models are correctly detected tomatoes. The IOU of four different models were 0, 0.913, 0.735 and 0.921 respectively. It reveals that SSD_Inception V2 model has the best performance.

Because the growth posture of cherry tomatoes is uncontrollable, the different postures of cherry tomatoes are roughly divided into two categories: front-grown and side-grown. The cherry tomatoes growing on the front are shown in Figure 5. In this state, the characteristics of cherry tomatoes are complete. However, only half of the number of fruit can be seen in side-grown cherry tomatoes. As shown in Figure 8, side-grown and front-grown tomatoes have the same length but the varying width. The shape features of side-grown are not exactly same as those of the front-growth. So traditional methods of only using shape features cannot primely detect tomatoes. The performance of the proposed methods were evaluated using 50 bunches side-grown cherry tomatoes. The results are shown in Table 5. The precision on the test set under side-grown condition was respectively of 30%, 86%, 76%, 74%. Some examples are shown in Figure 8. It also shows that the three models except SSD300 can correctly detect cherry tomatoes. Due to larger input image size of network, SSD512 behaved best in detecting small objects; in particular, some cherry tomatoes located at the edge of the image with only part of the fruit will be detected as a whole bunch of fruit.

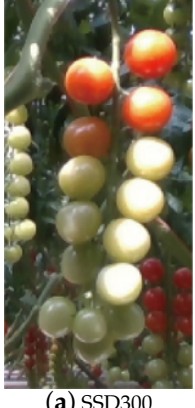 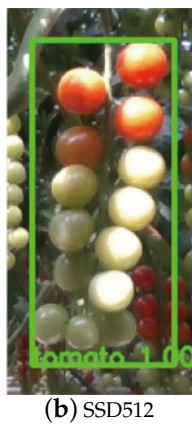 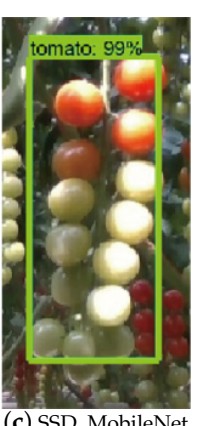 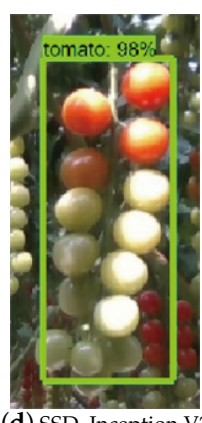

(**a**) SSD300　　　　　　(**b**) SSD512　　　　　　(**c**) SSD_MobileNet　　　　　(**d**) SSD_Inception V2

**Figure 7.** The detection results of uneven illumination cherry tomatoes. (**a**) the result of SSD300 model, (**b**) the result of SSD512 model, (**c**) the result of SSD_MobileNet model and (**d**) the result of SSD_Inception V2 model.

**Table 5.** The detection results of different methods under side-grown condition.

| Conditions | Methods | Amount | Correctly Identified Amount | P | Missed Amount | F |
|---|---|---|---|---|---|---|
| Side-grown | SSD300 | 50 | 15 | 30% | 35 | 70% |
| | SSD512 | | 43 | 86% | 7 | 14% |
| | SSD_MobileNet | | 38 | 76% | 12 | 24% |
| | SSD_Inception V2 | | 37 | 74% | 13 | 26% |

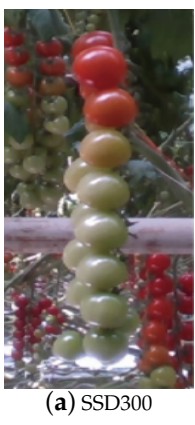 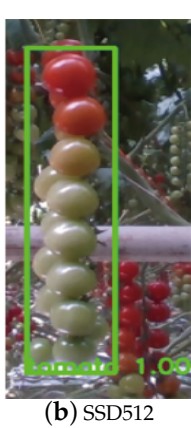 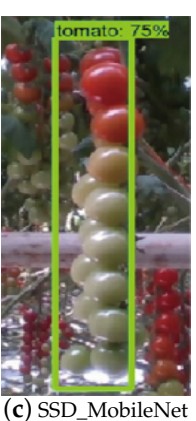 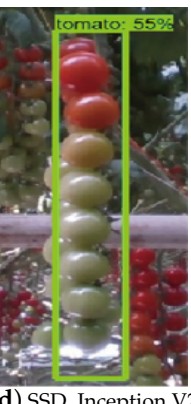

(**a**) SSD300　　　　　　(**b**) SSD512　　　　　　(**c**) SSD_MobileNet　　　　　(**d**) SSD_Inception V2

**Figure 8.** The detection results of side-grown cherry tomatoes. (**a**) the result of SSD300 model, (**b**) the result of SSD512 model, (**c**) the result of SSD_MobileNet model and (**d**) the result of SSD_Inception V2 model.

For cherry tomatoes under overlapped conditions, there are considerable overlapping areas, posing challenges to tomato detection. Figure 9 shows two bunches of cherry tomatoes numbered I ,II from front to back.Tomatoes I is easy to detect due to the complete characteristics, but tomatoes II was almost half covered. In Figure 9d, two tomatoes were both correctly detected, while only tomato I was correctly detected in Figure 9a–c. As for robotic picking, all cherry tomatoes identified at one shoot will considerably reduce the picking time.

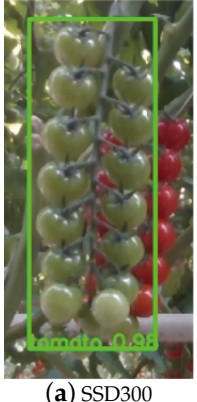 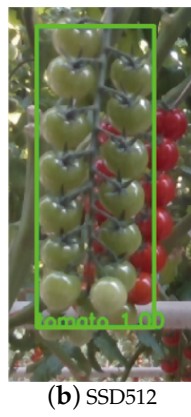 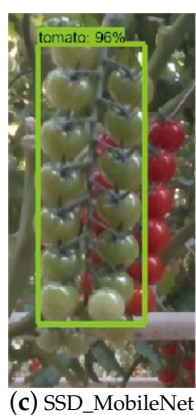 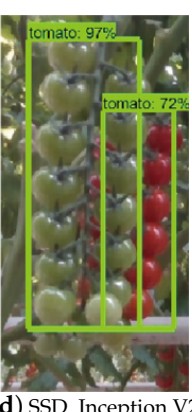

(**a**) SSD300          (**b**) SSD512          (**c**) SSD_MobileNet          (**d**) SSD_Inception V2

**Figure 9.** The detection results of overlapped cherry tomatoes. (**a**) the result of SSD300 model, (**b**) the result of SSD512 model, (**c**) the result of SSD_MobileNet model and (**d**) the result of SSD_Inception V2 model.

As shown in Figure 10, there are three bunches of cherry tomatoes in the image, numbered with I, II, and III from left to right. The tomatoes I presents the condition of immature tomatoes with light spots blocked by the main stem. The tomatoes II which shows the condition of side-grown is close to the tomatoes I. The tomatoes III shows the condition of half-ripe tomatoes with large-area light spots blocked by the main stem. Besides, there is an interference of another cherry tomatoes on the right side of tomatoes III.

As seen in Figure 10a, the SSD300 model failed to detect tomatoes. Although there are four prediction boxes in Figure 10b, just two of them were correctly detected. Some part of the tomatoes I was identified as a part of tomatoes II, so the output box of tomatoes II is larger than the ground truth box. That means, SSD512 model cannot separate those closing bunches. The tomatoes III was correctly detected but some part of the tomatoes III was mistaken for another tomatoes because of interference. Therefore, SSD512 model cannot provide an accurate picking position reference for automatic picking. Unlike SSD300 and SSD512, SSD_MobileNet correctly detects tomatoes III as shown in Figure 10c, but the output box of tomatoes II is larger than the ground truth box similarly. As shown in Figure 10d, tomatoes were correctly detected except tomatoes II.

Above all, it can be seen that SSD300 is not suitable for the conditions of uneven illumination, side-grown and closely, and SSD512 is sensitive to interference from other tomatoes and easily causes false detection. In addition, SSD_MobileNet easily causes false detection with relatively low rate of IOU compared with other three models under the separated conditions. Although in some conditions SSD_Inception V2 is not the model with the highest detection accuracy rate, it is the model with the lowest rate of false detection. As for uneven lighting, side-grown and nearly conditions, SSD_Inception V2 is more suitable to provide backstopping for automatic picking and avoid the problem of picking failure caused by false detection.

Table 6 shows the AP values of different SSD network models. Based on the above analysis, it can be observed that the accuracy of SSD300 model is relatively lower than others, and the performance of conditions such as uneven illumination, occlusion and overlapped, etc. are not effective. SSD512 has a great advantage in the detection of side-grown tomatoes. However, SSD512 is easy to cause false detection by the interference of adjacent tomato bunches. SSD_MobileNet is similar to SSD512 in false detection problem. Although the location loss is much less than SSD512, it still causes some damages to the fruit in the process of picking. SSD_Inception V2 with the detection accuracy of the lowest false detection rate as shown in Figure 10. If the results of SSD_Inception V2 are used as the reference position will greatly reduce the rate of fruit damage and the probability of repeated detection in the same location. Also it is very robust to different conditions of cherry tomatoes in greenhouse environment. It is shows that SSD_Inception V2 model had the highest accuracy, which

demonstrated that the method was effective and could be applied for the detection of cherry tomatoes in greenhouse environment.

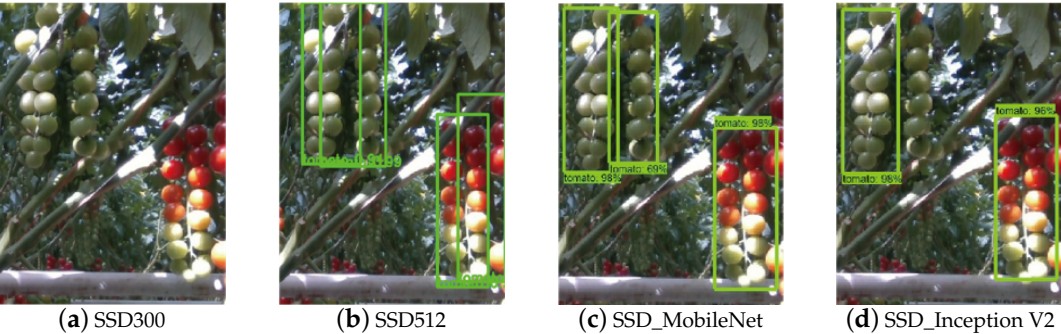

| **(a)** SSD300 | **(b)** SSD512 | **(c)** SSD_MobileNet | **(d)** SSD_Inception V2 |

**Figure 10.** The detection results of lateral growth cherry tomatoes. (**a**) the result of SSD300 model, (**b**) the result of SSD512 model, (**c**) the result of SSD_MobileNet model and (**d**) the result of SSD_Inception V2 model.

**Table 6.** Comparison test of different model.

| Evaluation Index | Different Model | | | |
|:---:|:---:|:---:|:---:|:---:|
| | SSD300 | SSD512 | SSD_MobileNet | SSD_Inception V2 |
| AP/% | 92.73 | 93.87 | 97.98 | 98.85 |

## 5. Conclusions and Future Work

In this paper, an improved cherry tomatoes detection method based on the classic SSD model in a greenhouse environment was proposed. Ripe tomatoes had a significant difference in color with background. It was easy to segment ripe tomatoes and background in the ideal state. In practice, however, because of uncertainty such as illumination, it was hard to segment objects with color and shape features in images. Compared with conventional detection methods only using color and shape features [27,28], this method can automatically extract features. In addition, this method does not only reduce the influence of the illumination, growth states and occlusion factors, but also improved the successful detection rate by 15.35% compared with conventional detection method. Moreover, this method to a great extent satisfied the practical application and greatly increased the success rate of robotic harvesting operation.

Through the analysis of the experimental results, it was found that the Inception V2 network was the best feature extractor of the SSD network for cherry tomatoes detection compared with VGG16 network and MobileNet network. The AP of SSD_Inception V2 reaches 98.85%. The correct identification rate, moreover, is the highest among four models. This model is 6.43% more accurate than classic SSD model.The correct identification rate was 94.00% for separated cherry tomatoes, 92% for occlusion, 96% for sunny or uneven illumination, 98% for shadow and 74% for side-grown cherry tomatoes. In addition, the main causes of cherry tomatoes misidentified are due to the interferences of the main stems and side-grown. Although the correct identification rate is not the highest in side-grown condition, it is not easy to cause false detection.Compared with other methods, SSD_Inception V2 performed best in all experiments. This model was proposed in this paper can be applied to cherry tomatoes recognition in greenhouse environment. It solves the problem that conventional recognition algorithms cannot recognize the fruit under changing conditions, and provide powerful information for robotic harvesting operation.

However, there are still some problems in the proposed method. The accuracy is not satisfactory for the side-grown tomatoes. In this condition, the rate of missed detection is high, therefore, our further work will focus on improving accuracy to the side-grown tomatoes. Moreover, our further work will

also include practical applications of harvesting robot equipped with the detection method proposed in this paper in greenhouse environment.

**Author Contributions:** Conceptualization, T.Y. and W.L.; methodology, L.L.; software, L.L. and F.Z.; validation, J.Z., C.Z. and W.Z.; formal analysis, F.Z.; investigation, L.L. and J.F.; resources, W.L.; data curation, L.L.; writing–original draft preparation, L.L.; writing–review and editing, T.Y. and F.Z., J.G.; visualization, L.L.; supervision, J.Z.; project administration, T.Y.; funding acquisition, W.L. All authors have read and agree to the published version of the manuscript.

**Funding:** This research was funded by "Research supported by the National Key Research and Development Program of China" OF 2016YFD0701501. The APC was funded by China Agricultural University.

**Acknowledgments:** The authors would like to give thanks to Hongfu Agricultural Tomato Production Park in Daxing for the expeimental images.

**Conflicts of Interest:** The authors declare no conflict of interest.

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
