# Peer review of "Robust Cherry Tomatoes Detection Algorithm in Greenhouse Scene Based on SSD"

_agriculture, doi:10.3390/agriculture10050160_

Round 1

Reviewer 1 Report

The main aim of the paper is to analyse a method based on deep learning for cherry tomatoes detection to reduce the influence of illumination, growth difference and occlusion.

The paper shows interesting information that can be useful for growers and designers. However, comparison with other studies is lacking and also a deeper discussion of the results and its applications. Why a practical applications was not done? Any result of that? The last part of the paper should be rewritten focussing on comparisons and applications

Specific comments

Line 19. Better to use square m than square hm. Better Mg than tons

Line 50. An explanation of SSD is needed as it is principal for the paper. It is explained deeper later but some explanation is needed in the introduction.

Line 83. If there were 1,617 images and they were modified with methods of rotating, brightness adjustment, and noising, why only were produced 3,330 images? Not all the methods were applied to all images? Which criteria was followed?

Line 190. How positive samples were determined and false positive? Images were classified by someone with experience? Which criteria was followed?

Line 202. Discussion is lacking. Comparison with other studies in tomato or other crops or systems of detection.

Line 286. Practical applications?

Reviewer 2 Report

Please find attached my suggestions and comments:

The Introduction section should be re-written, appears too simple in some part and sentences appears disconnected from each other. Too many citations in the introduction section without any critical discussion.

The Conclusion section should be improved, the conclusions need to be more to the point, it should be a precise reflection on key findings, what it means for practice and identification of key gaps and outlook.

English should be improved, I suggest a mother-tongue revision.

Round 2

Reviewer 1 Report

The main aim of the paper is to analyse a method based on deep learning for cherry tomatoes detection to reduce the influence of illumination, growth difference and occlusion.

The paper shows interesting information that can be useful for growers and designers. However, comparison with other studies is lacking and also a deeper discussion of the results and its applications. Why a practical applications was not done? Any result of that? The last part of the paper should be rewritten focussing on comparisons and applications. Improvements have been made, but still some aspects should be corrected.

Specific comments

Line 19. Better Mg than tons. Even the authors state that are common units they are not part of the international unit system. There are different kind of tons (metric, large, short) so it is ambiguous and international metric system should be used in stead of those units.

Line 83. If there were 1,617 images and they were modified with methods of rotating, brightness adjustment, and noising, why only were produced 3,330 images? Not all the methods were applied to all images? Which criteria was followed? The authors state that only some images were modified, but which criteria was used to modify some images and not others? Why not modify all the images and have more data no check the algorithm.

Line 190. How positive samples were determined and false positive? Images were classified by someone with experience? Which criteria was followed? The authors answered in the coverletter, but that information does not appear in the paper.

Line 202. Discussion is lacking. Comparison with other studies in tomato or other crops or systems of detection. The authors improved conclusions but the discussions still lacks of a proper comparison with the results of other studies.

Line 286. Practical applications? Still some improvements should be made. It’s economic compared with hand hasrvest? There is a big investment to be made by farmers? Will be used?
